# Direct Current Stimulation over the Primary Motor Cortex, Cerebellum, and Spinal Cord to Modulate Balance Performance: A Randomized Placebo-Controlled Trial

**DOI:** 10.3390/bioengineering11040353

**Published:** 2024-04-04

**Authors:** Jitka Veldema, Teni Steingräber, Leon von Grönheim, Jana Wienecke, Rieke Regel, Thomas Schack, Christoph Schütz

**Affiliations:** 1Faculty of Psychology and Sports Science, Bielefeld University, 33615 Bielefeld, Germany; teni.unciyan@uni-bielefeld.de (T.S.); leonvongroenheim@gmx.de (L.v.G.); rieke.regel@uni-bielefeld.de (R.R.); thomas.schack@uni-bielefeld.de (T.S.); christoph.schuetz@uni-bielefeld.de (C.S.); 2Department of Exercise and Health, Paderborn University, 33098 Paderborn, Germany; wienecke@sportmed.uni-paderborn.de

**Keywords:** tDCS, tsDCS, balance, postural control, primary motor cortex, cerebellum, spinal cord, healthy people

## Abstract

Objectives: Existing applications of non-invasive brain stimulation in the modulation of balance ability are focused on the primary motor cortex (M1). It is conceivable that other brain and spinal cord areas may be comparable or more promising targets in this regard. This study compares transcranial direct current stimulation (tDCS) over (i) the M1, (ii) the cerebellum, and (iii) trans-spinal direct current stimulation (tsDCS) in the modulation of balance ability. Methods: Forty-two sports students were randomized in this placebo-controlled study. Twenty minutes of anodal 1.5 mA t/tsDCS over (i) the M1, (ii) the cerebellum, and (iii) the spinal cord, as well as (iv) sham tDCS were applied to each subject. The Y Balance Test, Single Leg Landing Test, and Single Leg Squat Test were performed prior to and after each intervention. Results: The Y Balance Test showed significant improvement after real stimulation of each region compared to sham stimulation. While tsDCS supported the balance ability of both legs, M1 and cerebellar tDCS supported right leg stand only. No significant differences were found in the Single Leg Landing Test and the Single Leg Squat Test. Conclusions: Our data encourage the application of DCS over the cerebellum and spinal cord (in addition to the M1 region) in supporting balance control. Future research should investigate and compare the effects of different stimulation protocols (anodal or cathodal direct current stimulation (DCS), alternating current stimulation (ACS), high-definition DCS/ACS, closed-loop ACS) over these regions in healthy people and examine the potential of these approaches in the neurorehabilitation.

## 1. Introduction

Non-invasive DCS is a powerful tool modulating neural processing and can be successfully used for research and therapies. DCS consists of the application of a low-intensity direct current that flows between two or more electrodes. Present data indicate that a single session of tDCS can induce neurophysiological changes up to 120 min beyond the stimulation period [1,2,3], and its persistence increases linearly with the duration and the intensity of current applied [1,2]. A simplified theory distinguishes between anodal tDCS (with anode placed over the region of interest and cathode over another cranial or extracranial region) and cathodal tDCS (with reverse electrode positioning). Anodal tDCS should induce depolarization of neurons and increase corticospinal excitability. In contrast, cathodal stimulation should lead to a hyperpolarization of neurons and decrease corticospinal excitability [4,5,6]. Indeed, the real data show a large variability outside of this theoretical scope [7]. A key factor that determines the tDCS-induced effects is electrode positioning. A current systematic review indicates that tDCS applied over different regions modulates different aspects of walking in healthy people. While application over the primary motor cortex (M1) and cerebellum improved speed, synchronization, and variability during simple walking, dorsolateral prefrontal cortex (DLPFC) stimulation improved gait parameters under dual-task conditions [8]. However, another systematic review points to the fact that diverse interactions exist between tDCS specifications (M1/cerebellum, unilateral/bilateral/central, single/multiple sessions) and motor task interactions (uni/bi-manual, greater/less difficulty) [9]. This makes it difficult to draw clear conclusions. In addition, the reference electrode positioning may significantly impact the tDCS-induced effects. A simulation study (based on a numerical body model) compared six different cathode positions (right temporal lobe, right supraorbital region, right deltoid, left deltoid, under the chin, and right buccinator muscle) during anodal tDCS over the left M1 [10]. The results indicate that extracephalic electrodes may be more effective in the modulation of the spinal cord and similar or less effective in the modulation of the brainstem, than cephalic electrodes [10]. Another modeling study shows that a multipolar tDCS, with two anodes (over the right and the left M1) and one cathode (either over the spinal cord or over the right deltoid) may be effective in the modulation of deep brain structures, such as the thalamus, midbrain, and brainstem [11]. Numerous authors suggest that tsDCS, with one electrode over the spinal cord and the other electrode over another extracephalic region (such as spinal cord, deltoid muscle, iliac crest, etc.) are promising alternatives to conventional cranial applications [12]. Our study extends the knowledge on this field and investigates (in a direct comparison) the effects of three different electrode placements in modulating balance ability.

Balance and postural control are complex sensorimotor functions controlled by integrated brain and spinal networks [13,14,15]. Their neural background is still not fully understood. A recent systematic review with a meta-analysis emphasized the key role of the brainstem, cerebellum, basal ganglia, thalamus, and several cortical regions based on (functional) magnetic resonance imaging ((f) MRI) and positron emission tomography (PET) data [13]. Similarly, another systematic review indicated the key role of the cerebellum and brainstem, followed by the basal ganglia, thalamus, hippocampus, inferior parietal cortex, and frontal lobe regions, using MRI investigations [14]. Additionally, the spinal cord seems to play a crucial role in balance and postural control, as indicated by electrophysiological studies [15]. It has been repeatedly demonstrated that balance training leads (in addition to an improved balance ability) to spinal adaptations in the form of a suppressed Hoffmann reflex (H-reflex) [15,16].

Although the available data indicate that several cortical and subcortical brain regions, the cerebellum, and the spinal cord are crucially involved during motor control [13,14,15], the present applications of DCS focus mainly on M1 [17,18,19]. The evidence for the remaining central and peripheral nervous system is insufficient, similar to studies that directly compare the stimulation over different areas [17,18,19]. Therefore, the question arises whether other regions may be comparable or even more promising for DCS applications. Our study investigates and compares the effectiveness of t/tsDCS over the M1, cerebellum, and spinal cord [20,21,22].

## 2. Methods

### 2.1. Study Design

This was a randomized placebo-controlled crossover study. Three single sessions of real t/tsDCS (over the (i) M1, (ii) cerebellum, and (iii) spinal cord) and one session of sham tDCS were applied to each participant in a randomized order (PC-generated) with a washout period of at least 48 h in between. Balance ability was evaluated immediately before and after each intervention. The study was conducted according to the standards established by the Declaration of Helsinki, approved by the Ethics Committee of Bielefeld University (2022-043), and entered in the German Clinical Trial Register on 28 September 2023 (DRKS00032749). 

### 2.2. Participants

The inclusion criteria were as follows: (1) age between 18 and 25 years, (2) no contraindications for tDCS (checked by safety screening questionnaire [23]), and (3) no relevant neurological, psychiatric, or orthopedic disorders. All subjects provided their written informed consent prior to participation. A G*power analysis (effect size = 0.25, α error probability *p* < 0.05, Power = 0.95) revealed that a sample size of at least 40 participants is needed to detect statistically significant effects using ANOVA with four interventions and two timepoints.

### 2.3. Intervention

Each subject completed four separate 20 min interventional sessions: (1) 1.5 mA tDCS over the M1, (2) 1.5 mA tDCS over the cerebellum, (3) 1.5 mA tsDCS over the spinal cord, and (4) sham tDCS (stimulator turned off after 5 s) over M1. A DC-stimulator PLUS (NeuroConn Gmbh, Ilmenau, Germany) and two saline-soaked sponge electrodes (5 cm × 7 cm) were used. For M1 stimulation, the anode was placed over the Cz, and the cathode was placed over the right supraorbital area (Fp2). For cerebellar stimulation, the anode was placed over the O2, and the cathode was placed over the right buccinator muscle. The electrodes positioning for M1 and cerebellar tDCS is in line with previous studies [24,25]. For spinal stimulation, the anode was placed over the spinal cord at the Th8 level, and the cathode was placed over L2. A simulation study indicated that this electrode placement is superior (in comparison to deltoid, umbilicus, and iliac crest cathode placements) regarding the electric field generated in lumbar and sacral spinal segments [26]. The international 10/20 EEG system [27] and palpation method [28,29] were used to determine electrode positioning during M1, cerebellar and spinal t/tsDCS. Figure 1 shows the electrodes’ placements used in this study.

### 2.4. Assessments

Three different assessments (Y Balance Test, the Single Leg Landing Test, and the Single Leg Squat Balance Test) were used to evaluate balance ability. The right and the left leg were tested in a randomized order during each test. The investigators were blinded to intervention allocation.

The Y Balance Test was performed using a test kit (FMS, Chatham, VA, USA). The maximal reach of the free lower leg in the (a) anterior, (b) posterolateral, and (c) posteromedial directions was determined during a one leg stance on the opposite leg [30]. A better balance ability was associated with a greater reach distance. Five trials were performed for each leg and direction. The mean value of the five trials was used for analysis.

During the Single Leg Landing Test, participants were instructed to perform a forward jump (50% of their body height), land on a single limb, and achieve a stable position as quickly as possible [31,32,33]. The center of gravity (COG) in the anterior–posterior and medial–lateral directions and the time taken to regain balance were recorded using a force plate (AMTI, Watertown, MA, USA). A smaller COG area and a faster time to stabilize indicated better balance. Five trials were performed for each leg. The mean value was used for the analysis.

During the Single Leg Squat Test, probands performed five consecutive single-leg squats (10% of their body height) [31,34]. The center of gravity (COG) in the anterior–posterior and medial–lateral directions was recorded using the force plate described above. The smaller the COG area was, the better the balance. Two trials were performed for each leg. The mean values were used for the analysis.

### 2.5. Analysis

The SPSS software package, version 27 (International Business Machines Corporation Systems, IBM, Ehningen, BW, Germany), was used to analyze the data collected during this study. The independent sample *t*-tests evaluated pre-interventional comparability. Repeated-measure ANOVAs with the factors “intervention” and “time” compared the pre–post changes across interventions. Mauchly’s sphericity tests and Greenhouse–Geisser corrections were applied. Due to multiple comparisons, a *p*-value of ≤0.01 was considered statistically significant. The outliers (mean ± 3 SD) were excluded from the analysis. The researcher performing statistical analysis was not blind to intervention allocation.

## 3. Results

Overall, 42 participants were randomized (age 25.1 ± 3.2 years, 19 females, 23 males, 36 right-footed, and 6 left-footed). The foot preferred to kick the ball was considered to be dominant [35]. All participants tolerated the interventions well without severe adverse events. Four participants reported less severe side effects, such as a burning sensation and nausea (one participant after M1 stimulation) and a metallic taste in the mouth (three participants after cerebellar stimulation). The pre-interventional data did not differ significantly across interventions. Table 1 summarizes the data on balance collected during the experiment. The outliers (4% of values) were removed. The ANOVAs detected significant time*intervention interactions on the Y Balance Test, but not on the Single Leg Landing Test and the Single Leg Squat Test. The effects were observed more frequently for the left leg than for the right leg. For the left leg, a significant improvement of balance ability (in comparison to the sham tDCS) was detected after M1 (F_1,40_ = 8.999; *p* = 0.005) (F_1,36_ = 18.624; *p* < 0.001), cerebellar (F_1,40_ = 8.796; *p* = 0.005) (F_1,36_ = 16.291; *p ≤* 0.001) and spinal (F_1,39_ = 13.55; *p ≤* 0.001) (F_1,34_ = 8.799; *p* = 0.005) application for the posterior-lateral and posterior-medial directions, respectively. For the right leg, only tsDCS induced significantly greater effects than the sham tDCS for both the posterior-lateral (F_1,39_ = 11.53; *p* = 0.002) and posterior-medial (F_1,39_ = 7.943; *p* = 0.008) directions. No significant effects were observed for the anterior direction. The intervention-induced effects did not significantly differ across real t/tsDCS interventions. Figure 2 and Figure 3 illustrate the intervention-induced changes.

## 4. Discussion

The aim of this study is to investigate and compare the effects of 1.5 mA t/tsDCS applied over the M1, cerebellum, and spinal cord on balance and postural control. The data show that (1) stimulation of each region significantly improved balance and postural control during the Y Balance Test but not during the Single Leg Landing Test and the Single Leg Squat Test, and (2) spinal stimulation improved the balance ability of both legs, while M1 and cerebellar stimulation improved the right leg stand only.

### 4.1. Stimulated Area Specific Modulation

Although several neuroimaging data indicate that several cortical and subcortical regions, the cerebellum, the brainstem, and the spinal cord, are crucially involved during balance and postural control [13,14,15], the majority of existing studies have applied tDCS over the M1 [17,18,19]. The previous evidence for the remaining regions was insufficient. Direct comparisons of different regions regarding t/tsDCS-induced effects on balance and postural control were almost non-existent [17,18,19]. We have demonstrated that the cerebellum and spinal cord are promising targets for the application of t/tsDCS in supporting balance control, in addition to M1. Thus, our results provide an important contribution to this field. Accordingly, a review suggests that the core systems of the automatic process of postural control are mostly achieved by the brainstem and spinal cord, while the forebrain structures and cerebellum act on the brainstem–spinal cord systems so that the cognitive processes of postural control can be achieved [36]. A model developed in the 1990s indicated that so-called central pattern generators (CPGs) could play a crucial role in gait and posture control [37,38,39]. CPGs are located in the lower thoracic and lumbar regions of the vertebrate spinal cord and drive rhythmic and stereotyped motor behavior such as walking or swimming without input from higher brain areas [37,38,39]. It is assumed that spinal reflex networks are crucially involved in these self-organizing neural circuits [40,41]. This finding is supported by studies that detected the suppression of H-reflexes after balance training, in parallel to balance and postural control improvement [15,16]. Besides this, it is cogitable that the orientation of neurons within the spinal cord (highly orientated axons extending along the craniocaudal axis) [42] leads to more consistent tsDCS-induced effects in comparison to cerebral tDCS application (inconsistent axons extending within the gyral banks) [43,44].

### 4.2. Leg-Specific Modulation

Our data show a greater improvement in the balance ability for standing on the left leg than on the right leg. This is true for M1 and cerebral tDCS, but not for tsDCS. This can be caused by electrode positioning in relation to the sagittal body plane in our study. The electrodes were placed symmetrically during tsDCS (anode over Th8 and cathode over L2). In contrast, a stronger right-hemispheric modulation was expected from M1 tDCS (anode over Cz and cathode over right supraorbital area) [45] and cerebellar tDCS (anode over O2 and cathode over right buccinator muscle) [46]. Indeed, the effects detected in our study are not consistent with the theory that the cerebrum controls the contralateral hemi body and the cerebellum controls the ipsilateral hemi body [47,48]. This theory (among others) is based on fMRI investigations that show that the active movement of a single lower limb is associated with increased neural activation of the primary sensorimotor cortex, supplementary motor area, cingulate motor area, secondary somatosensory cortex, and basal ganglia of the contralateral hemisphere, but with increased neural activation within the ipsilateral anterior lobe of the cerebellum [47].

A growing number of studies have demonstrated hemispheric asymmetries of motor control [47,49,50]. FMRI data show that brain activation during a movement of the non-dominant limb is more bilateral than during the same movement performed with the dominant extremity [47]. A TMS study demonstrated that the voluntary movement of a hand resulted in an increase in MEP amplitude in the non-task hand. This increase was more pronounced during left hand movements than during left hand tasks [50]. Accordingly, lesion studies indicate that the non-affected hemisphere can compensate for damage to the non-dominant hemisphere rather than for damage to the dominant hemisphere [49,51]. Hand motor recovery after left hemispheric stroke is two to three times slower than that after a right hemispheric incident [49,51]. Thus, one may assume that non invasive brain stimulation (NIBS) over the dominant hemisphere has the potential to modulate neural processing and/or motor control within the whole body, while the targeting of the non-dominant hemisphere modulates the non-dominant hemi body only. Unfortunately, there exists insufficient evidence in this regard. Existing NIBS research strongly focuses on the dominant hemisphere (and leg) and neglects the non-dominant hemisphere (and extremities). Future research should address this gap.

### 4.3. Balance Task-Specific Modulation

Our data demonstrate that the choice of assessment significantly influences the effects. While numerous significant t/tsDCS-induced improvements were found on the Y Balance Test, no effects were detected on the Single Leg Landing Test and the Single Leg Squat. Accordingly, numerous studies have shown little consistency in balance performance when using different assessments [52,53]. Balance performance during (i) single leg landing, (ii) stance on unstable platform, and (iii) forward falls correlated only weakly in young healthy people [53]. Similarly, performance during (i) the bipedal stance, (ii) stance on unstable platform, and (iii) the Functional Reach Test were not correlated in children aged 7–10 years [52]. It can be assumed that different mechanisms are responsible for balance control. An interesting perspective on this topic offers the Balance Evaluation Systems Test [54]. This test battery differentiates between six balance control systems (biomechanical constraints, stability limits/verticality, anticipatory postural adjustments, postural responses, sensory orientation, and stability in gait). This assessment was developed to identify the underlying cause for poor functional balance in several cohorts [54]. Participants with balance deficiencies in one category do not necessarily show deficits in other categories [54]. Future studies should closely evaluate the relationships between balance performance measured by differential assessments and the neural processing within brain and spinal cord networks. Besides this, the relationships between deep and superficial sensitivity and balance and/or gait performance should be investigated. The existing data show that balance and postural control are complex neural processes, and their neural background has not been fully understood to date. 

## 5. Strengths and Limitations

This is the first placebo-controlled study that compared the effects of t/tsDCS over the different areas of balance ability in healthy participants. Our results provide additional insights into the neural background of balance and postural control and support the development of innovative therapy strategies in several cohorts. A weakness of our experiments is the limited number of participants; missing neuro-navigation to determine the exact location of optimal tDCS application; and differential electrode positioning in relation to the sagittal body plane: (1) both electrodes over the midline during tsDCS, (2) anode over the midline and cathode over the right supraorbital area during M1 tDCS, and (3) anode over the right cerebellum and cathode over the right buccinator muscle during cerebellar stimulation.

## 6. Conclusions

Our study indicates that both cerebellum and spinal cord are promising areas for the application of NIBS in supporting balance ability in young healthy people. While tsDCS supported balance in both legs, M1 and cerebellar tDCS supported only right leg balance. It is an open question whether, and to what extent, our finding can be transferred to people of different ages and disabled populations. Future research should fill the gap of evidence on this field and investigate the effects of tDCS applications over different brain and spinal cord regions in different cohorts. The incorporation of neuro-navigation to precisely target stimulation areas can improve the specificity and efficacy of the results. The tsDCS should be more closely examined in the framework of motor rehabilitation. The motor disabilities caused by several neurological diseases are associated not only with changes in neural processing within the brain but also within the spinal networks [55,56,57,58]. E.g., an increased spinal reflex is observed in stroke victims compared to healthy people and its normalization correlates with a successful gait recovery [55,56]. A desynchronization of spinal reflex loop oscillations seems to be one of the main sources of tremor in Parkinson’s disease [57]. An increasing number of dystonia cases (traditionally considered a disorder of the basal ganglia, brainstem, and cerebellum) are reported in patients with spinal cord pathology [58]. tsDCS can be a promising alternative to “traditional” cerebral applications in these cohorts [55,56,57,58].

## Figures and Tables

**Figure 1 bioengineering-11-00353-f001:**
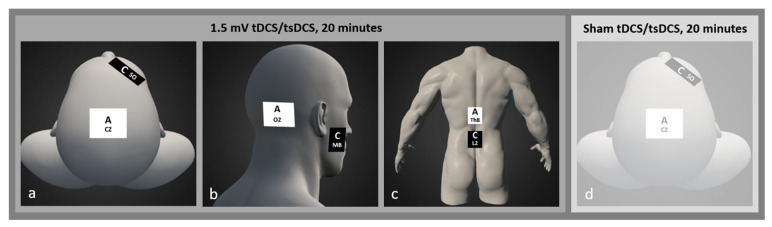
Electrodes positioning used for (**a**) M1 tDCS, (**b**) cerebellar tDCS, (**c**) spinal tDCS and (**d**) sham tDCS.

**Figure 2 bioengineering-11-00353-f002:**
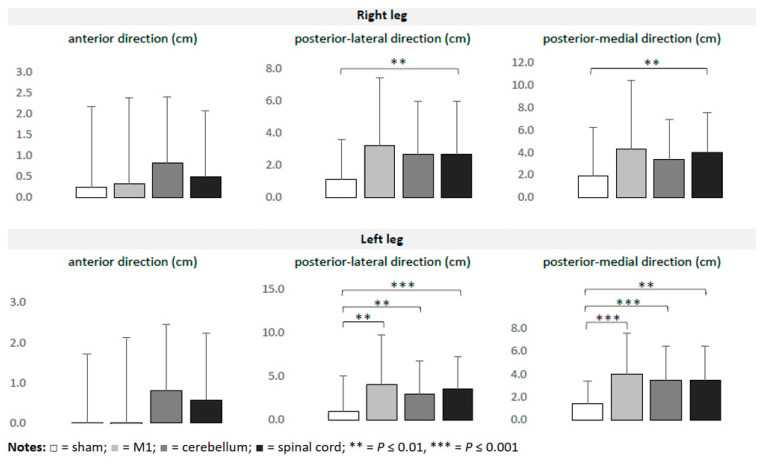
Intervention-induced changes (means and SD) in the Y Balance Test in relation to baseline.

**Figure 3 bioengineering-11-00353-f003:**
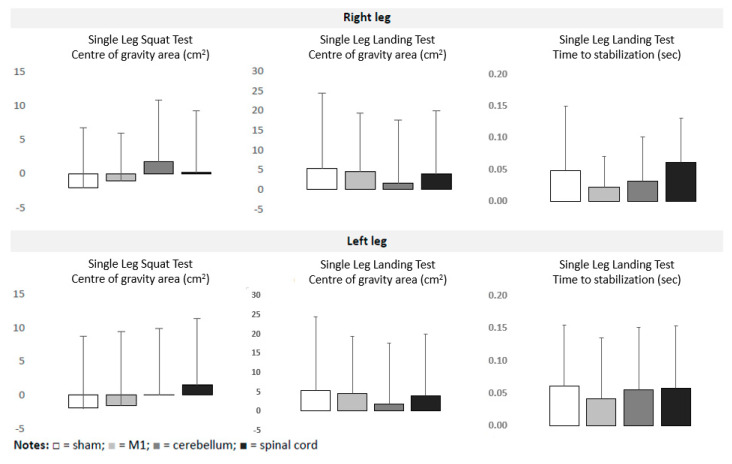
Intervention-induced changes (means and SD) in Single Leg Squat Test and Single Leg Landing Test in relation to baseline. Notes: 

 = sham; 

 = M1; 

 = cerebellum; 

 = spinal.

**Table 1 bioengineering-11-00353-t001:** Balance performance (means and SD) at both time-points (pre, post).

				Sham tDCS	M1 tDCS	Cerebellar tDCS	Spinal tDCS
Y Balance Test	Right leg	Anterior direction (cm)	pre	57.29 ± 7.41	56.10 ± 5.57	57.54 ± 7.45	56.42 ± 5.31
			post	57.53 ± 7.80	56.43 ± 5.80	58.36 ± 7.76	56.91 ± 5.54
		Posterolateral direction (cm)	pre	105.21 ± 12.07	103.72 ± 10.54	104.45 ± 12.76	104.00 ± 11.09
			post	106.26 ± 12.47	106.48 ± 12.31	107.14 ± 13.35	107.06 ± 12.47 **
		Posteromedial direction (cm)	pre	101.19 ± 12.76	99.18 ± 13.42	102.17 ± 13.99	99.47 ± 11.23
			post	103.03 ± 13.05	103.48 ± 14.42	105.53 ± 14.03	103.45 ± 11.70 **
	Left leg	Anterior direction (cm)	pre	57.76 ± 7.02	56.96 ± 5.77	56.98 ± 5.87	57.02 ± 4.85
			post	57.75 ± 7.08	56.92 ± 5.57	57.77 ± 5.92	57.60 ± 5.27
		Posterolateral direction (cm)	pre	103.78 ± 11.17	102.50 ± 10.41	103.58 ±± 12.24	102.54 ± 10.14
			post	104.68 ± 11.12	106.54 ± 12.50 **	106.56 ± 12.74 **	106.06 ± 11.24 ***
		Posteromedial direction (cm)	pre	101.58 ± 11.15	99.88 ± 13.66	101.48 ± 14.02	100.30 ± 11.15
			post	102.99 ± 11.12	103.9 ± 14.11 ***	104.98 ± 14.27 ***	103.75 ± 10.65 **
Single Leg Landing Test	Right leg	Center of gravity area (mm^2^)	pre	5163 ± 1486	5363 ± 1470	5363 ± 1678	5406 ± 1690
			post	5619 ± 1975	5530 ± 1665	5752 ± 1910	5938 ± 1764
		Time to stabilization (ms)	pre	1.196 ± 0.180	1.211 ± 0.185	1.141 ± 0.154	1.191 ± 0.154
			post	1.219 ± 0.188	1.243 ± 0.206	1.203 ± 0.147	1.240 ± 0.160
	Left leg	Center of gravity area (mm^2^)	pre	5437 ± 1224	5257 ± 1017	5017 ± 1173	5305 ± 1282
			post	5997 ± 1580	5154 ± 1281	5574 ± 1546	6133 ± 1501
		Time to stabilization (ms)	pre	1.232 ± 0.153	1.273 ± 0.177	1.227 ± 0.187	1.264 ± 0.170
			post	1.274 ± 0.196	1.329 ± 0.192	1.285 ± 0.210	1.326 ± 0.161
Single Leg Squat Test	Right leg	Center of gravity area (mm^2^)	pre	3020 ± 1246	2967 ± 1312	3006 ± 988	3232 ± 1057
			post	2950 ± 1071	3147 ± 1089	3085 ± 1201	3043 ± 1073
	Left leg	Center of gravity area (mm^2^)	pre	3241 ± 1274	3223 ± 1159	3179 ± 987	3298 ± 973
			post	3087 ± 1299	3223 ± 1302	3329 ± 817	3093 ± 898

Notes: intervention-induced changes in comparison to sham ** = *p* ≤ 0.01; *** = *p* ≤ 0.001.

## Data Availability

The datasets generated during and/or analyzed during the current study are available from the corresponding author on reasonable request.

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
