# Peer review of "Direct Current Stimulation over the Primary Motor Cortex, Cerebellum, and Spinal Cord to Modulate Balance Performance: A Randomized Placebo-Controlled Trial"

_bioengineering, 2024, doi:10.3390/bioengineering11040353_

Round 1

Reviewer 1 Report

Comments and Suggestions for Authors

You could consider adding a few additional paragraphs to elaborate on the findings and their clinical significance.

Explicitly mention the rationale behind exploring regions beyond the M1 for balance modulation, or the potential impact on clinical practices and rehabilitation outcomes.

Detailed parameters of the tDCS and tsDCS protocols used, including the specific current intensity, duration, and electrode placement?

What are the precise models of the equipment and materials used?

Participant Details: More comprehensive demographic data on the participants, including age range, gender distribution, and any other relevant characteristics, such as physical fitness level or history with balance-related activities.

Neuro-navigation: The absence of neuro-navigation to determine exact tDCS electrode placement is identified as a limitation. Future studies could incorporate neuro-navigation to precisely target stimulation areas, which could improve the specificity and efficacy of the results.

Stimulation Parameters: Detailed descriptions of the tDCS/tsDCS protocols should be included, such as exact current strength, duration of the stimulation, number of sessions, and electrode size and type. 

Balance Tests: Clear, step-by-step protocols for the Y Balance Test, Single Leg Landing Test, and Single Leg Squat Test should be described, including how the baseline and post-intervention measurements were taken.

Statistical Analysis: Description of statistical tests used, including justification for their use, details on the handling of any missing or outlying data, and the level of statistical significance considered.

Randomization and Blinding: The process for randomizing participants to different intervention groups should be elaborated, along with any blinding methods used to prevent bias from participants or investigators.

Electrode Positioning: A rationale for the chosen electrode placements, along with diagrams or images, might help other researchers replicate the setup more precisely.

Control Measures: Explanation of handling potential confounding variables and controls put in place to minimize their effects.

Ethical Considerations: Details about the process for obtaining ethics approval and informed consent could be expanded upon to ensure ethical considerations are clear.

Limitation Discussion: A critical discussion of the limitations of the study's methodology, including how these limitations could affect the results and how they might be addressed in future research.

How did you decide that the statistical analysis is robust and appropriate for the study design?

If the participant pool is not diverse, concisely discuss how this might affect the generalizability of the results and the potential implications for different populations.

Consider and discuss alternate interpretations of the data or results that contradicted expectations, providing a balanced view.

Author Response

Dear reviewer, thank you very much for taking the time to review our manuscript. We revised our paper in accordance with your comments (changes are marked in red) and have included a point-by-point response.

Reviewer 2 Report

Comments and Suggestions for Authors

I was very pleased to be introduced to such an interesting study. The study focuses on the problems of supporting the ability to balance. The results obtained may be useful for clinical physiology. Thus, the proposed method promotes the application of DCS on cerebellum and spinal cord to support balance control. The study design is adequate, materials and methods are described correctly. I have no conceptual comments. The results are presented clearly and support the discussion.

Author Response

Dear reviewer, thank you very much for the positive review of our manuscript.